# Rapid detection of *ERG11* polymorphism associated azole resistance in *Candida tropicalis*

**Saikat Paul, Rajneesh Dadwal** **, Shreya Singh, Dipika Shaw, Arunaloke Chakrabarti, Shivaprakash M. Rudramurthy, Anup K. Ghosh** *

Department of Medical Microbiology, Postgraduate Institute of Medical Education and Research (PGIMER), Chandigarh, India

* anupkg3@gmail.com

**Data Availability Statement:** All relevant data are within the manuscript and its Supporting Information files.

## Abstract

Increasing reports of azole resistance in *Candida tropicalis*, highlight the development of rapid resistance detection techniques. Nonsynonymous mutations in the lanosterol C14 alpha-demethylase (*ERG11*) gene is one of the predominant mechanisms of azole resistance in *C. tropicalis*. We evaluated the tetra primer-amplification refractory mutation system-PCR (T-ARMS-PCR), restriction site mutation (RSM), and high-resolution melt (HRM) analysis methods for rapid resistance detection based on *ERG11* polymorphism in *C. tropicalis*. Twelve azole-resistant and 19 susceptible isolates of *C. tropicalis* were included. DNA sequencing of the isolates was performed to check the *ERG11* polymorphism status among resistant and susceptible isolates. Three approaches T-ARMS-PCR, RSM, and HRM were evaluated and validated for the rapid detection of *ERG11* mutation. The fluconazole MICs for the 12 resistant and 19 susceptible isolates were 32–256 mg/L and 0.5–1 mg/L, respectively. The resistant isolates showed A339T and C461T mutations in the *ERG11* gene. The T-ARMS-PCR and RSM approaches discriminated all the resistant and susceptible isolates, whereas HRM analysis differentiated all except one susceptible isolate. The sensitivity, specificity, analytical sensitivity, time, and cost of analysis suggests that these three methods can be utilized for the rapid detection of *ERG11* mutations in *C. tropicalis*. Additionally, an excellent concordance with DNA sequencing was noted for all three methods. The rapid, sensitive, and inexpensive T-ARMS-PCR, RSM, and HRM approaches are suitable for the detection of azole resistance based on *ERG11* polymorphism in *C. tropicalis* and can be implemented in clinical setups for batter patient management.

## Introduction

*Candida* species are common commensals residing on human skin, genitourinary, respiratory, and gastrointestinal tracts. However, they also hold pathogenic potential causing a wide range of infections ranging from superficial to serious life-threatening systemic disease [1, 2]. Invasive candidiasis(IC) is most commonly seen in immunocompromised patients and is

**Funding:** We have completed the study as a part of a PhD thesis and by utilizing the institutional research grant (No. 71/2-Edu-16/4856 Dated: 12/ 12/2017, Budget allotment of Rs. 4,75,000/-). Additionally, the manpower for this study was supported by the Indian Council of Medical Research (ICMR), Government of India.

**Competing interests:** The authors have declared that no competing interests exist.

associated with high morbidity and mortality [3–5]. Among the non-*Candida albicans Candida* (NCAC) species *Candida tropicalis* is the first common cause of candidemia in African countries like Tunisia and Algeria [6, 7]. In Asian countries including India, *C. tropicalis* is reported to be one of the most predominant yeast causing IC particularly in elderly, immuno-compromised patients and those in critical care settings [2, 4, 8, 9].

Triazoles are the most commonly used antifungal agents for the treatment of IC in developing countries, where the high expenses of echinocandins deter their wide use in such countries [8, 9]. They act by inhibiting the enzyme lanosterol C14 alpha-demethylase (*Erg11p*), an important component of the fungal ergosterol biosynthesis pathway encoded by the *ERG11* gene. Several studies have reported the emergence of azole resistance in *C. tropicalis* [2–4, 8, 9]. This could be associated with several drug-related, host, and pathogen-associated factors including the misuse of antifungal drugs, inappropriate duration of antifungal therapy, lack of restrictions on the use of drugs in agriculture and horticulture industries, etc. [3, 8]. Additionally, azole-resistant *C. tropicalis* isolates may even occur in azole-naive patients that might suggest horizontal transfer in clinical settings [10–12].

Despite the multitude of mechanisms described till date, amino acid alterations due to the mutations in the coding sequence in the *ERG11* gene is perhaps the most important mechanism behind azole resistance in *C. tropicalis* [13–16]. Among the nonsynonymous mutations in the *ERG11* gene, A395T and C461T are the most frequently reported in resistant isolates [13–17]. Although C461T mutation does not confer azole resistance, it commonly appears along with A395T [17]. Therefore, these two mutations could be used as important markers of azole resistance detection in *C. tropicalis*.

The Clinical and Laboratory Standards Institute (CLSI) and the European Committee on Antimicrobial Susceptibility Testing (EUCAST) provide guidelines for antifungal susceptibility testing (AFST) and routinely used in the clinical setup. However, there are some limitations including long turn-around time, tedious procedures and subjective interpretation of results [18–23]. Therefore, alternative approaches are crucial for prompt and accurate documentation of high minimum inhibitory concentration (MIC) or antifungal resistance to ensure appropriate therapy. In *C. tropicalis*, mutations in the *ERG11* gene are one of the predominant mechanisms of azole resistance and detection of these mutations is performed by DNA sequencing [13–17]. Despite attempts at other approaches, sequencing remains the gold standard for mutation detection but is unfortunately time-consuming and expensive [24]. Given the rising azole resistance in *C. tropicalis*, the development of alternative molecular approaches are imperative for rapid, reliable, accurate, and cost-effective detection of various *ERG11* mutations for optimum selection of antifungal therapy to aid patient management.

Therefore, in the present study, we have developed and evaluated the tetra primer-amplification refractory mutation system-PCR (T-ARMS-PCR), restriction site mutation (RSM) and high-resolution melt (HRM) analysis approaches for rapid detection of *ERG11* mutations in the clinical isolates of *C. tropicalis*.

## Materials and methods

### Isolates and growth conditions

*C. tropicalis* isolates causing IC were screened from 2015 to 2018 and the azole-resistant isolates were deposited to the National Culture Collection of Pathogenic Fungi (NCCPF), Postgraduate Institute of Medical Education and Research (PGIMER), Chandigarh, India. In the present study, a total of 31 isolates (12 azole-resistant and 19 susceptible) were used. This study was approved by the Institute ethics committee PGIMER, Chandigarh, India. The isolates were grown on Sabouraud's dextrose agar with chloramphenicol (HiMedia, India) and

incubated for 24 hours at 37˚C. Matrix assisted laser desorption ionization-time of flight mass spectrometry [(MALDI-TOF MS); Microflex LT mass spectrometer, Bruker Daltonik, Bremen, Germany)] and DNA sequencing of the internal transcribed spacer (ITS) region was utilized for the identification of the isolates [25, 26].

### *In-vitro* antifungal susceptibility testing

CLSI recommended M27-A3 and M27-S4 guidelines for the broth microdilution (BMD) was followed for the assessment minimum inhibitory concentrations (MICs) against fluconazole, voriconazole, itraconazole, and posaconazole [18, 19].

### Sequencing of *ERG11* gene

Overlapping primers were designed by using the NCBI Primer-BLAST tool and the complete coding sequence of the *ERG11* was amplified for sequencing as described in our previous study (S1 Table in S1 File) [27]. The *ERG11* gene sequence of *C. tropicalis* MYA-3404 was used as a reference for primer designing and mutation analysis. The complete coding sequence of the *ERG11* gene from all the isolates was submitted to the NCBI GenBank and the isolate specific accession numbers are presented in Table 2 and S2 Table in S1 File.

### T-ARMS-PCR approach

Primers for T-ARMS-PCR were designed by using the web-based primer designing platform Primer 1 (http://primer1.soton.ac.uk/primer1.html) to assess the most frequently noticed A395T and C461T mutations in the *ERG11* gene (Table 1) [28, 29]. NCBI Primer-BLAST tool (https://www.ncbi.nlm.nih.gov/tools/primer-blast/) was used to assess the specificity of the designed primers. The PCR amplification was performed in a 20 µL reaction volume containing 100 ng DNA, 1x PCR buffer with MgCl$_2$, 0.2 mM dNTPs, 0.5 µM of each allele-specific and outer primers and 1 U of Taq polymerase (GeNei, India). PCR program for the thermal cycler (Eppendorf, Germany) was as follows: an initial denaturation for 5 minutes at 95˚C, followed by 35 cycles of 1 minute at 94˚C, 30 seconds primers annealing at 60˚C, 1minute amplification at 72˚C and the final extension step of 7 minutes at 72˚C. The amplified products were subjected to 2% agarose gel electrophoresis at 400 mAmp and 110 V for 30–45 minutes. The amplified products were analyzed under UV in a gel documentation instrument (Alpha Innotech, California) [29–31]. We also examined the detection limit of this approach by using 100, 10, 1, 0.1, and 0.01 ng of DNA input.

### RSM assay for mutation detection

Web-based NCBI Primer-BLAST tool was used to construct the primers for RSM assay and the *ERG11* gene sequence from *C. tropicalis* MYA-3404 was used as a reference (Table 1). The quality of the primers was examined by the web-based software Sequence Manipulation Suite (www.bioinformatics.org). The PCR amplification was performed as described above except the primers used (0.5 µM of each forward and reverse primers). We could not find any restriction sites for A395T and HinfI restriction enzyme (New England Biolabs, USA) specific for the 'GANTC' sequence was used to detect the C461T mutation. Restriction digestion was performed in 25 µL reaction volume containing 20 µL amplified product, 2.5 µL 10X NEBuffer, 1 µL HinfI restriction enzyme, 1.5 µL milli-Q water and the reaction mixture was incubated at 37˚C for 15 minutes [32]. The digested products were separated and visualized as described earlier. The limit of detection was also evaluated as previously mentioned.

**Table 1. The details of the primers used for T-ARMS-PCR, RSM, and HRM analysis.**

| | Mutations | Sequence (5′->3′) forward and reverse | Ta | Product (bp) |
|---|---|---|---|---|
| **T-ARMS-PCR** | A395T | FOP: TAGCATGGCAATTACTTTACTCCTTA | 60˚C | Outer primers: 474 |
| | | ROP: GTTGAGTTTTCATAACACTAGCAACAC | | A allele: 212 |
| | | A allele: ACTCCTGTTTTTGGTAAAGGTGTTATATA | | T allele: 318 |
| | | T allele: CCATTAATCTAGAGTTTGGACAATGAA | | |
| | C461T | FOP: AAAGATAGAGTTCCAATGGTTTTCTACTGG | 60˚C | Outer primers: 536 |
| | | ROP: TCAGCATACAATTGAGCAAATGATCT | | C allele: 237 |
| | | C allele: TTTGCTAAATTTGCTTTGACTACTGAGTC | | T allele: 355 |
| | | T allele: TGATCTTTGGAACATAGGTTTTGACAA | | |
| **RSM** | C461T | FP: TCTACTGGATCCCATGGTTTGG | 60˚C | Amplicon: 571 |
| | | RP: TGAGGTAATGGCAAGTTTGGG | | |
| **HRM** | A395T & C461T | FP: ACTCCTGTTTTTGGTAAAGGTGT | 60˚C | Amplicon: 131 |
| | | RP: ACTTCTTCTCTGATCTTTGGAACA | | |

FOP: Forward outer primer; ROP: Reverse outer primer, FP: Forward outer primer; RP: Reverse primer; Ta: Annealing Temperature, bp: Base pairs

## HRM for mutation screening

A primer pair was designed including both the A395T and C461T mutations in the *ERG11* gene by using the MYA-3404 reference sequence (Table 1). The HRM assay was performed on the LightCycler 480 (Roche, Switzerland) with the Kapa HRM Fast Kit (Merck, USA). The reaction was performed in a total volume of 20 μL. 1 μL of 100 ng/μL DNA was added to a reaction mixture containing 10 μl 2X Kapa HRM Fast Mastermix (Merck, USA), 0.5μM final concentration of each primer (Sigma-Aldrich, Germany) and milli-Q water. The PCR thermo-cycling conditions were as follows: initial denaturation at 95˚C for 3 minutes, 50 cycles with denaturation at 95˚C for 5 seconds and annealing/extension at 60˚C for 25 seconds followed by the HRM ramping from 65˚C to 95˚C. Fluorescence data were acquired at 0.02˚C increments every 1 second to generate amplicon specific melting curves. Data analysis was performed using Roche system software (Roche, Switzerland), normalized and difference plots were generated to visualize the differences in the amplicons [33].

**Table 2. Clinical details, MIC distribution, and mutation status of the azole resistant isolates.**

| NCCPF ID | GenBank accession number | Source of isolates | Flu MIC (mg/L) | Vori MIC (mg/L) | Itra MIC (mg/L) | Posa MICs (mg/L) | *ERG11* mutations | Amino acid alterations |
|---|---|---|---|---|---|---|---|---|
| 420189 | MW015956 | Blood | 128 | 4 | 0.5 | 0.5 | A395T & C461T | Y132F & S154F |
| 420227 | MW015957 | Pus | 128 | 0.5 | 0.25 | 0.5 | A395T & C461T | Y132F & S154F |
| 420232 | MW015958 | Blood | 32 | 0.5 | 0.5 | 0.5 | A395T & C461T | Y132F & S154F |
| 420233 | MW015959 | Blood | 32 | 1 | 0.25 | 0.25 | A395T & C461T | Y132F & S154F |
| 420234 | MW015960 | Blood | 64 | 1 | 0.25 | 0.25 | A395T & C461T | Y132F & S154F |
| 420235 | MW015961 | Blood | 32 | 0.5 | 0.25 | 0.25 | A395T & C461T | Y132F & S154F |
| 420236 | MW015962 | Blood | 32 | 0.5 | 0.25 | 0.25 | A395T & C461T | Y132F & S154F |
| 420237 | MW015963 | Blood | 64 | 1 | 0.5 | 0.5 | A395T & C461T | Y132F & S154F |
| 420238 | MW015964 | Ascitic fluid | 256 | 16 | 16 | 2 | A395T & C461T | Y132F & S154F |
| 420239 | MW015965 | Blood | 256 | 16 | 16 | 0.5 | A395T & C461T | Y132F & S154F |
| 420245 | MW015966 | Blood | 128 | 4 | 1 | 0.5 | A395T & C461T | Y132F & S154F |
| 420247 | MW015967 | Wound slough | 128 | 4 | 2 | 0.25 | A395T & C461T | Y132F & S154F |

Flu: Fluconazole; Vori: Voriconazole; Itra: Itraconazole; Posa: Posaconazole; Y: Tyrosine; F: Phenylalanine; S: Serine

## Results

### Details of the isolates used

Twelve fluconazole-resistant isolates (MIC range: 32–256 mg/L) with A339T and C461T mutations and 19 susceptible isolates (MIC range: 0.5–1 mg/L) without these mutations were obtained from NCCPF to include in the present study (Table 2 and S2 Table in S1 File). Of the 12 fluconazole-resistant isolates, 8 were voriconazole resistant (1–16 mg/L), 4 itraconazole (4–16 mg/L) and 1 posaconazole resistant (2mg/L). Two resistant and susceptible isolates each were used for the initial standardization of T-ARMS-PCR, RSM, and HRM approaches and further validated by using the rest of the isolates.

### Mutation detection by T-ARMS-PCR approach

Fig 1A representing the schematic diagram of the amplified products for the detection of A395T mutation in the *ERG11* gene. The Forward outer primer (FOP) and Reverse outer primer (ROP) for the A395T transition produce a 474 base pair (bp) fragment. In resistant isolates, the mutated 'T' sequence-specific inner reverse (IR) primer and FOP produced a 318 bp fragment. While, the 'A' sequence-specific inner forward (IF) primer and ROP produced a 212 bp fragment in case of susceptible isolates (Fig 1B and S1 and S2 Figs in S1 File). The reliability and reproducibility of the T-ARMS-PCR approach were confirmed by putting up the reactions in triplicate. The analytical sensitivity of this method was examined by diluting the input DNA concentration and it was noted that the T-ARMS-PCR method could detect up to 10 ng of DNA sample (Fig 1C and S1 Fig in S1 File).

Similarly, for the detection of C461T mutations, IF and ROP produced a 237 bp product for susceptible isolates. While IR and FOP produced a 355 bp product for resistant isolates (Fig 2A and 2B and S3 and S4 Figs in S1 File). The analytical sensitivity was the same (10 ng) as

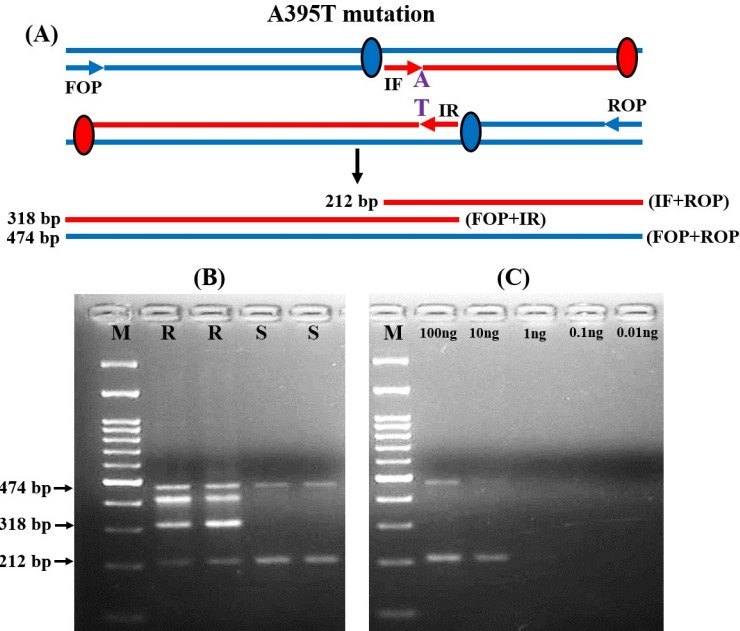

**Fig 1. T-ARMS-PCR analysis of *ERG11* gene mutation among resistant (R) and susceptible (S) isolates.** (A) Schematic representation of T-ARMS-PCR assay for A395T alteration. (B) Representative agarose gel electrophoresis of the T-ARMS-PCR assay amplicons for both R and S isolates with and without *ERG11* mutations. (C) Analytical sensitivity of T-ARMS-PCR examined by diluting the DNA. M: 100 bp molecular weight markers.

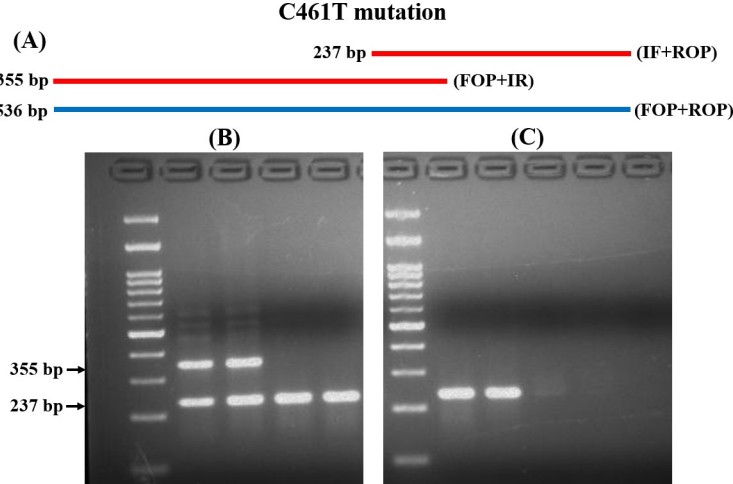

**Fig 2. T-ARMS-PCR analysis of C461T mutation in *ERG11*.** (A) Schematic diagram of T-ARMS-PCR for C461T alteration. (B) Representative gel image of the fragment produced in R and S isolates. (C) Analytical sensitivity of T-ARMS-PCR examined by diluting the DNA. M: 100 bp molecular weight markers.

previously described (Fig 2C and S3 Fig in S1 File). The T-ARMS-PCR approach accurately discriminated all the resistant and susceptible isolates.

## RSM assay for mutation screening

The most important determinant of RMS is the presence of the target sequence of a restriction enzyme at the mutation site. In the present study, we could not find any restriction enzyme specific site for the detection of A395T mutation. Thus, we standardized this method for the detection of C461T mutation, an equally significant contributing mutation for azole resistance, by using the HinfI restriction enzyme. The forward primer (FP) and reverse primer (RP) specific for the upstream and downstream region of C461T mutation amplified all the isolates and produced a 571 bp sized product. In resistant isolates, the amplified product with the 'GANT**T**' sequence at the 461 position could not be cleaved by HinfI and the product length remained the same. Whereas, HinfI enzyme cleaved the 'GANT**C**' sequence and produced 268 and 306 bp fragments for susceptible isolates (Fig 3A). After restriction digestion, the resistant and susceptible isolates specific fragments are presented in Fig 3B and S5 and S6 Figs in S1 File. We also examined the analytical sensitivity of the RMS approach by diluting the template DNA and it was noted to be up to 1 ng (Fig 3C and S5 Fig in S1 File). The RMS analysis correctly differentiated the resistant and susceptible isolates.

## Screening of *ERG11* mutation by HRM assay

The HRM analysis differentiates the amplified products that have single nucleotide polymorphism by generating different types of melting curves during the time of heating after amplification. Hence, we adopted the HRM approach for the rapid detection of *ERG11* mutations in *C. tropicalis*. A single primer set covering both 395 and 461 regions were used for screening the mutational resistance. The normalized melting curve showed two variants, indicating variations in *ERG11* gene sequences among resistant and susceptible isolates (Fig 4A and S7 Fig in S1 File). The difference plot has also confirmed the results of the melting curve analysis (Fig 4B and S8 Fig in S1 File).

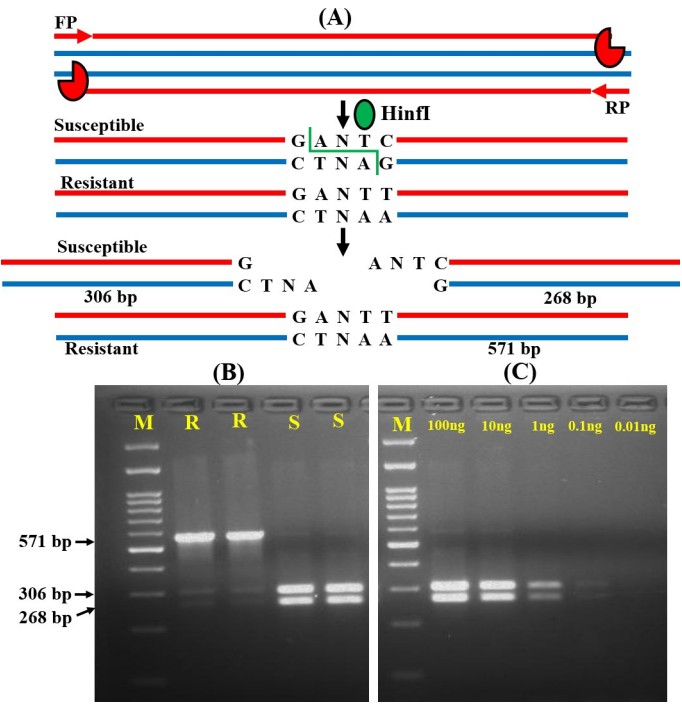

**Fig 3. RSM analysis for *ERG11* mutation screening.** (A) Schematic representation of RSM assay for the C461T mutation screening among resistant (R) and susceptible (S) isolates. (B) Agarose gel image of the fragments specific for R and S isolates (C) Gel image of gradually diluted DNA samples to confirm the analytical sensitivity of the RSM assay. M: 100 bp molecular weight markers.

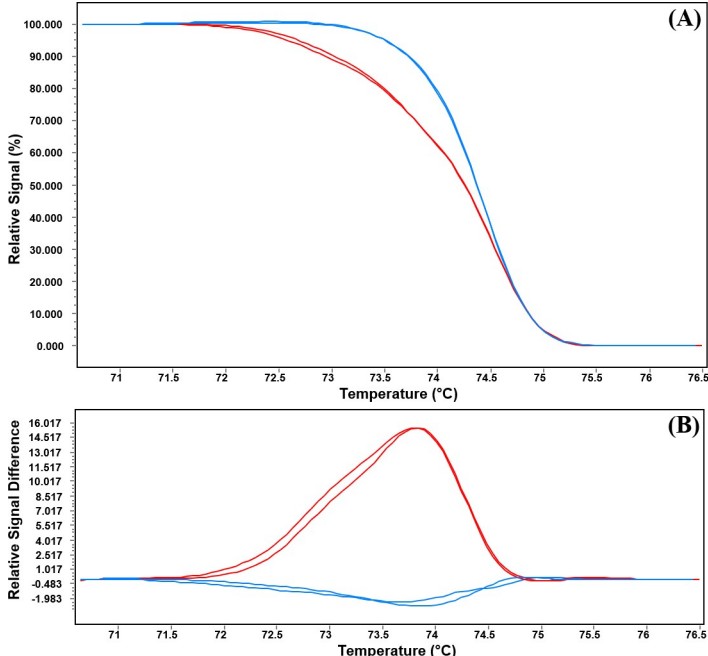

**Fig 4. HRM analysis of the *ERG11* gene of *C. tropicalis*.** (A) Normalized melting curve and (B) Difference plot presenting two variants of the *ERG11* gene fragment among the resistant and susceptible isolates. Red curves resistant variant and blue curves susceptible variant.

The melting temperature (Tm) analysis confirmed the presence of molecular alterations between resistant and susceptible isolates. The Tm for resistant isolates with *ERG11* mutations was 74.16±0.06, significantly different from the Tm susceptible isolates (74.55±0.11) with wild type sequence (p<0.0001) (Fig 5). Only one susceptible isolate was noted to present Tm of 74.25, similar to that of the resistant isolates.

## Comparison of standardized approaches with DNA sequencing

A comparative analysis was performed to determine the suitability of the approaches for the rapid detection of *ERG11* mutations in *C. tropicalis*. We compared the methods with respect to the sensitivity, specificity, time required for detection, cost of analysis, and detection limit in our setup (Table 3). All these developed approaches were suitable for the rapid detection of resistance based on *ERG11* mutations in *C. tropicalis*.

## Discussion

Azoles are commonly used for the treatment of infections due to *Candida* species [8, 9]. With the increasing reports of azole resistance in *C. tropicalis*, an understanding of the mechanisms of resistance and development of rapid, reliable and robust resistance detection methods is crucial [2–4, 8, 9]. Various factors may contribute to the development of azole resistance in *C. tropicalis* [13–17]. Of these, mutations in the coding sequence of *ERG11* are directly related to the significant escalations of resistance against different azole antifungal drugs in clinical settings [2, 13–17, 34]. Therefore, detection of *ERG11* mutation related resistance reliably and efficiently deals with resistance related issues in clinical setups.

The rapid detection of resistance to various antifungal drugs (azoles, echinocandins, and terbinafine) has been reported in many medically important fungi based on mutations in different resistance-related genes [30, 35–38]. Resistance detection in many yeasts and molds has already been performed by using allele-specific real-time molecular probes, DNA microarray, HRM analysis, real-time PCR with molecular beacon probes, pyrosequencing, PCR-restriction fragment length polymorphism (PCR-RFLP), fluorescence resonance energy transfer (FRET),

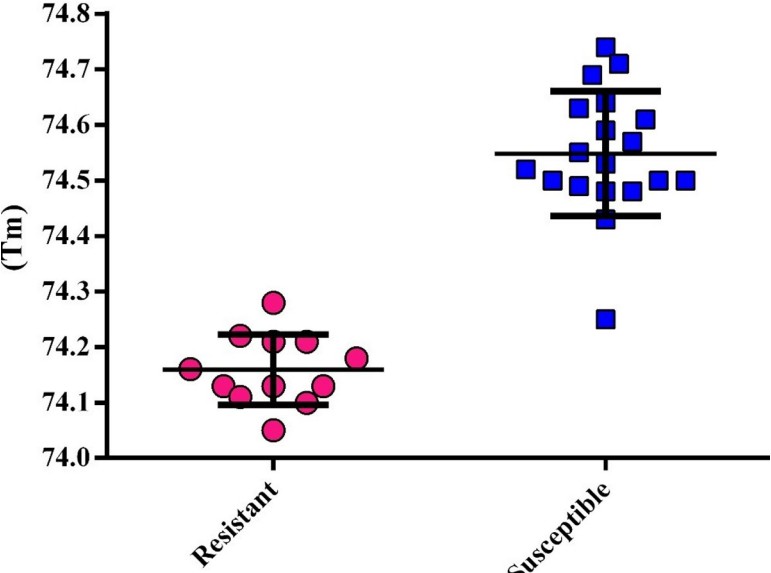

**Fig 5. The scatter dot plot representing the Tm distributions among resistant and susceptible isolates.**

**Table 3. Comparison between DNA sequencing, T-ARMS-PCR, RSM, and HRM approaches.**

|  | DNA sequencing | T-ARMS-PCR | RSM* | HRM |
|---|---|---|---|---|
| Sensitivity | 100% | 100% | 100% | 100% |
| Specificity | 100% | 100% | 100% | 94.74% |
| Detection time | ~24 hours | ~4 hours | ~5 hours | ~3 hours |
| Cost/reaction | ~15 US dollars | <1 US dollars | ~2 US dollars | <1 US dollars |
| Detection limit | 5 ng | 10 ng | 1 ng | 0.1 ng |

* Only for C461T mutation

rolling circle amplification (RCA), ARMS-PCR, etc. [30, 35–38]. Although studies have been performed on the rapid detection of *ERG11* mutations associated with azole resistance in other *Candida* species, *C. tropicalis* has not been explored yet [33, 38, 39]. In the present study, we demonstrate the excellent capability of three simple, rapid (<5 hours), cost-effective (<2 US dollars), and highly sensitive T-ARMS-PCR, RSM, and HRM-based approaches for the surveillance or detection of the most commonly reported A395T and C461T mutations in *ERG11* gene among the clinical isolates of *C. tropicalis* for the first time.

T-ARMS-PCR is an efficient approach used for SNP genotyping [29, 31, 40, 41]. A study from our clinical setup has been performed for the rapid terbinafine resistance detection in *Trichophyton* species by using conventional ARMS-PCR [30]. In conventional ARMS-PCR, the wild and mutant type alleles are amplified in two independent PCR reactions, in contrast, T-ARMS-PCR amplifies both alleles along with the control fragment together [29, 31]. Thus, we adopted T-ARMS-PCR for the first time to rapidly detect the *ERG11* mutations in *C. tropicalis*. Among the four primers used for T-ARMS-PCR, two wild and mutant allele-specific primers (IF and IR) were constructed in opposite directions with the combination of two outer primers (FOP and ROP) for the amplification of both the alleles simultaneously. As the designed primers produce different lengths of allele-specific amplicons with a significant size difference, they can be easily distinguished in agarose gel electrophoresis [29]. In the present study, T-ARMS-PCR differentiated all the resistant and susceptible isolates with respect to the variations in the amplicon size. In a single reaction, T-ARMS-PCR produces amplicons specific to wild-type, heterozygotes, or homozygotes mutations [29, 31]. In our study, both the resistant and susceptible isolates were clearly differentiated based on wild type 'A' and mutant type 'T' allele-specific products. Studies have also reported the presence of some nonspecific amplification in T-ARMS-PCR analysis [29, 31, 41]. Likewise, one nonspecific amplicon was seen only for A395T mutation detection in resistant isolates even in the present study. Finally, T-ARMS-PCR is an efficient method for the rapid detection of *ERG11* mutations in clinical *C. tropicalis* isolates.

RSM assay has been developed for the detection of mutations present in the specific target DNA sequence of the restriction enzyme [42, 43]. To the best of our knowledge, the RSM approach has not been implemented yet for the rapid detection of resistance in fungi and present study is the first to examine the *ERG11* mutations in *C. tropicalis* isolates. The limitation of the RSM approach is its complete dependency on the presence of a restriction site [32]. Therefore, we were only able to detect the C461T mutation as we did not find any restriction enzyme site for A395T mutation. In the RSM assay, the FP and RP amplified the genomic DNA of both resistant and susceptible isolates and after restriction enzyme treatment, the amplicon size was same in the resistant isolates due to the absence of specifies restriction site, whereas susceptible isolates produced two smaller fragments of different lengths and were resolved in gel electrophoresis. The RSM approach is an easy and suitable method for the rapid detection of mutation if the mutation is present in the restriction site.

HRM analysis is a sensitive and precise approach used for the identification of different *Candida* species [44]. This sensitive technique can discriminate the amplified products with a single nucleotide variation by generating different types of melting curves after amplification [45]. The HRM analysis also used for the prediction of azole resistance in *C. albicans* by examining the *ERG11* polymorphisms [33]. Therefore, we standardized this technique for the rapid screening of azole resistance in *C. tropicalis*. Two different types of melting curves and melting temperatures were noted for the resistant and susceptible isolates confirming the presence of two sequence variants among these two groups of isolates. HRM successfully discriminated all the resistant and susceptible isolates except one susceptible isolate which showed similarity with resistant isolates. Several studies have been performed for the rapid detection of *ERG11* mutations by FRET, RCA, asymmetric PCR with molecular beacon (MB) based melting curve analysis, and bioluminometric pyrosequencing [37–39, 46]. Due to the less complex nature and high resolution of HRM analysis, we propose this approach as an efficient approach for the rapid resistance detection in *C. tropicalis*.

Although DNA sequencing is the gold standard for mutation detection, high turnaround time (~24 hours) and running cost (~15 US dollars) limits its application in routine clinical setups. We also compared the suitability of the developed T-ARMS-PCR, RSM, and HRM approaches with DNA sequencing. All three methods were significantly less time consuming and inexpensive compared to sequencing. The sensitivity and specificity of these three methods were comparable to DNA sequencing. The study population was not very big and this might be the reason for significantly higher sensitivity and specificity. The analytical sensitivity of the HRM approach was significantly higher (0.1 ng) in comparison with other methods (>1 ng). It should also be noted that T-ARMS-PCR and RSM approaches could be adopted by using a conventional thermal cycler in those setups where DNA sequencer and real-time PCR facilities are not available. Finally, all these three methods could be used as an alternative to DNA sequencing for the rapid detection of *ERG11* mutations in *C. tropicalis*.

Along with the advantages, some limitations are also present in these approaches. Although, both A395T and C461T mutations frequently occur together, isolated A395T mutation can also be seen [13, 14, 17]. This would create a challenge to HRM analysis since A395T mutations do not substantially influence the Tm, thereby limiting the use of this technique as a comprehensive surrogate marker. Though the sensitivity and specificity of HRM were high (>90%) in our study, a very subtle difference in the Tm values was seen between azole-resistant and azole-susceptible isolates, which might cause misidentification. The same limitation is also applicable to the RSM approach, as it cannot differentiate resistant isolates from susceptible isolates carrying only A395T mutation. In our setup, among the 32 resistant isolates of *C. tropicalis*, 12 (37.5%) resistant isolates presented with *ERG11* mutations and were included in the present study. Further studies with a large number of isolates are essential for the further validation of the developed approaches, specifically in those centres where *ERG11* mutations have been reported in more than 90% of the resistant isolates [17]. Apart from the A395T and C461T mutations, several other mutations in the *ERG11* gene have been reported among azole-resistant *C. tropicalis* isolates [16, 17, 47]. We only found A395T and C461T mutations among the resistant isolates and therefore only these were evaluated in the present study. Further studies with other fungal species resistant to azoles and other drug classes are crucial for batter patient management and infection control.

## Conclusions

In conclusion, we have developed the rapid, inexpensive, sensitive, and specific T-ARMS-PCR, RSM, and HRM based diagnostic platforms for the screening of *ERG11* mutations in *C*.

*tropicalis* and exhibited excellent concordance with DNA sequencing. These approaches hold promise as simple and robust for the detection of azole resistance and can be implemented in routine clinical laboratories for effective therapy and epidemiological surveillance.

## Supporting information

**S1 File.**
(DOCX)

## Acknowledgments

We express our gratitude to the Department of Medical Microbiology, PGIMER, Chandigarh for allowing us to conduct this study.

## Author Contributions

**Conceptualization:** Saikat Paul, Anup K. Ghosh.

**Data curation:** Saikat Paul, Rajneesh Dadwal, Dipika Shaw, Anup K. Ghosh.

**Formal analysis:** Saikat Paul, Rajneesh Dadwal, Dipika Shaw.

**Investigation:** Anup K. Ghosh.

**Methodology:** Saikat Paul, Rajneesh Dadwal, Dipika Shaw, Anup K. Ghosh.

**Resources:** Anup K. Ghosh.

**Software:** Saikat Paul.

**Supervision:** Anup K. Ghosh.

**Validation:** Saikat Paul, Rajneesh Dadwal, Dipika Shaw, Anup K. Ghosh.

**Writing – original draft:** Saikat Paul.

**Writing – review & editing:** Saikat Paul, Rajneesh Dadwal, Shreya Singh, Dipika Shaw, Arunaloke Chakrabarti, Shivaprakash M. Rudramurthy, Anup K. Ghosh.

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
