## [Decision Letter · Decision Letter 0]

16 Jul 2020

PONE-D-20-17635

Rapid detection of ERG11 polymorphism associated azole resistance in Candida tropicalis

PLOS ONE

Dear Dr. Ghosh,

Thank you for submitting your manuscript to PLOS ONE. After careful consideration, we feel that it has merit but does not fully meet PLOS ONE’s publication criteria as it currently stands. Therefore, we invite you to submit a revised version of the manuscript that addresses the points raised during the review process.

Two reviewers, who are experienced with C. tropicalis and antifungal resistance, have reviewed your manuscript. It is agreed that improved understanding of antifungal resistance in C. tropicalis is needed and this manuscript begins to address this issue. However, the reviewers and myself believe that more than two isolates need to be studied. Additionally, there are some methodological issues raised by both reviewers, but particularly reviewer #1 which need to be addressed.

We look forward to receiving your revised manuscript.

Kind regards,

Joy Sturtevant

Academic Editor

PLOS ONE

Journal Requirements:

2. We note that you are reporting an analysis of a microarray, next-generation sequencing, or deep sequencing data set.

PLOS requires that authors comply with field-specific standards for preparation, recording, and deposition of data in repositories appropriate to their field.

We require the accession numbers for ERG11 sequencing data.

Please upload these data to a stable, public repository (such as ArrayExpress, Gene Expression Omnibus (GEO), DNA Data Bank of Japan (DDBJ), NCBI GenBank, NCBI Sequence Read Archive, or EMBL Nucleotide Sequence Database (ENA)). In your revised cover letter, please provide the relevant accession numbers that may be used to access these data.

For a full list of recommended repositories, see http://journals.plos.org/plosone/s/data-availability#loc-omics or http://journals.plos.org/plosone/s/data-availability#loc-sequencing

'The authors duly acknowledge the Indian Council of Medical Research (ICMR), Government of India for financial supports.'

'To,

The Editor

PLOS ONE

Subject: Request for the waiver of article processing charge (APC) in your esteemed journal

Respected Sir/Madam

We are hereby submitting the manuscript entitled “Rapid detection of ERG11 polymorphism associated azole resistance in Candida tropicalis” for your kind consideration. With the increasing reports of azole resistance in Candida tropicalis, development of rapid resistance detection techniques is crucial. We are for the first time evaluated and validated the tetra primer-amplification refractory mutation system-PCR (T-ARMS-PCR), restriction site mutation (RSM), and high resolution melt (HRM) analysis approaches for rapid resistance detection based on ERG11 polymorphism in C. tropicalis. We have tried our best to make this study up to the mark for PLOS ONE publication. We are submitting a manuscript for the first time in your esteemed journal. We have completed the study as a part of a PhD thesis and by utilizing the institutional research grant (No. 71/2-Edu-16/4856 Dated: 12/12/2017. Due to the lack of sufficient funds, we are unable to pay the article processing charge (APC) at this point. Please provide us the APC waiver so that we can publish our work in this reputed journal.   

With warm regards

Dr. Anup k Ghosh

Corresponding author'

**Please include your amended statements within your cover letter; we will change the online submission form on your behalf.

Reviewers' comments:

Reviewer's Responses to Questions

**Comments to the Author**

1. Is the manuscript technically sound, and do the data support the conclusions?

Reviewer #1: Partly

Reviewer #2: Yes

2. Has the statistical analysis been performed appropriately and rigorously? 

Reviewer #1: N/A

Reviewer #2: N/A

3. Have the authors made all data underlying the findings in their manuscript fully available?

Reviewer #1: Yes

Reviewer #2: Yes

4. Is the manuscript presented in an intelligible fashion and written in standard English?

Reviewer #1: Yes

Reviewer #2: Yes

5. Review Comments to the Author

Reviewer #1: Indeed, C. tropicalis is among the most clinically important Candida species and invention of techniques capable of differentiating azole-resistant from azole-susceptible isolates will have clinical implications. Methodologically speaking, however, there are some drawbacks to this study, which are as follows,

1. C461T does not confer azole resistance and the inclusion of RSM is not necessary (doi: 10.1016/j.cmi.2018.11.007), since it only detects this mutation.

2. The HRM assay shows a very subtle difference concerning the Tm values of the PCR product obtained from azole-resistant and azole-susceptible isolates, which for sure will cause misidentification and even this slight difference is because of substitution of Cytosine to Thymidine, while the substitution of Adenosine to Thymidine per se will not be differentiated via HRM application. Therefore, these points will also undercut the applicability of HRM and RSM, but ARMS PCR, only for A395T will be important.

3. In light of these points, authors are encouraged to obtain more azole-resistant C. tropicalis isolates carrying Y132F and azole-susceptible isolates, prepare blinded test sets, and subject them to their ARMS PCR. This will establish the basis for the applicability of ARMS PCR to be used in developing countries, where ICs are mainly treated by azoles and also more advanced PCR machines, such as real-time PCR, might not be that available (doi: 10.3390/jof5040090). Try to use a systematic flow proving the efficacy of ARMS PCR and supplement your findings with ROC curve and well-arranged figures.

Line 59, please note that C. tropicalis is the first common cause of candidemia in Tunisia (doi: 10.3109/13693786.2010.493561) and also Algeria (doi: 10.1186/s13756-020-00710-z).

Lines 60-63. Please note that triazoles are more commonly used in developing countries, where the high expenses of echinocandins deter their wide use in such countries.

Lines 63-66. Please note that azole-resistance is not caused by host factors rather by the prolonged and/ or previous exposure with azoles and poor hand hygiene and infection control measures may further amplify the azole-resistant isolates in clinical settings. Authors may meant azole therapeutic failure. Also, please highlight the fact that azole-resistant C. tropicalis isolates may even occur in azole-naïve patients that might suggest horizontal transfer in clinical settings. Examples are, doi: 10.1007/s10156-012-0412-9, doi: 10.3201/eid2509.190520, doi.org/10.1093/mmy/myz124.

As mentioned above, please note that C461T does not confer azole resistance, which has been proved via heterologous expression analysis (doi: 10.1016/j.cmi.2018.11.007).

Lines 73-79. These statements are too strong and authors are encouraged to modify this, since there are some studies also have shown the otherwise (doi.org/10.1093/mmy/myz124) and not all azole-resistant isolates harbor accountable mutations in ERG11. Indeed, azole resistance, unlike echinocandin resistance, involves numerous players and their contribution in concert will cause azole resistance. Moreover, here authors have just focused on one mutation, Y132F, while other mutations, although with a lower prevalence, confer azole resistance, such as G464S/D, Y125F, P56S, etc.

It seems unclear why authors developed so many methods, while they could have focused on one method. I mean, this is not reflected in introduction.

Lines 108-115, please delete them and simply refer to a study. No need for such elaboration on DNA extraction.

Why authors used such huge quantity of DNA samples, 100ng DNA? Even using 1ng DNA any PCR reactions work perfect.

Reviewer #2: None. Please see my comments in the attachment.

And I have no competing interest that interferes with or could be perceived as potentially interfering with, a thorough and objective assessment of this manuscript.

6. PLOS authors have the option to publish the peer review history of their article (what does this mean?). If published, this will include your full peer review and any attached files.

Reviewer #1: No

Reviewer #2: No

---

## [Author Response · Author response to Decision Letter 0]

18 Sep 2020

Editor comments:

Comment: Two reviewers, who are experienced with C. tropicalis and antifungal resistance, have reviewed your manuscript. It is agreed that improved understanding of antifungal resistance in C. tropicalis is needed and this manuscript begins to address this issue. However, the reviewers and myself believe that more than two isolates need to be studied. Additionally, there are some methodological issues raised by both reviewers, but particularly reviewer #1 which need to be addressed.

Response: Thank you for considering our manuscript in your esteemed journal. We have tried our best to address all the issues raised by the Academic Editor and Reviewers. In the manuscript, initially we presented the results by using two representative isolates, but as per your suggestion more number of isolates have been tested and the results are presented in the Supporting Information file (S2, S4, S6, S7 and S8 Fig) in the revised manuscript. 

Comment: Please include the following items when submitting your revised manuscript:

Response: We are submitting the rebuttal letter containing each point raised by the academic editor and reviewers.

We are also sending one copy of the ‘Revised Manuscript with Track Changes’ and one clean copy of the revised version labeled as ‘Manuscript’. 

Comment: Response: Required modifications in the financial disclosure has included in the updated statement of the cover letter. 

Comment: Guidelines for resubmitting your figure files are available below the reviewer comments at the end of this letter.

Response: We are resubmitting the figure files according to the guidelines of this journal. 

Comment: Please ensure that your manuscript meets PLOS ONE's style requirements, including those for file naming.

Response: Our revised manuscript meets the style requirements of PLOS ONE. 

Comment: We note that you are reporting an analysis of a microarray, next-generation sequencing, or deep sequencing data set.

PLOS requires that authors comply with field-specific standards for preparation, recording, and deposition of data in repositories appropriate to their field.

We require the accession numbers for ERG11 sequencing data.

Please upload these data to a stable, public repository (such as ArrayExpress, Gene Expression Omnibus (GEO), DNA Data Bank of Japan (DDBJ), NCBI GenBank, NCBI Sequence Read Archive, or EMBL Nucleotide Sequence Database (ENA)). In your revised cover letter, please provide the relevant accession numbers that may be used to access these data.

Response: We agree with you. Our manuscript contains the DNA sequence data of the complete coding sequence of the lanosterol C14 alpha-demethylase (ERG11) gene. As per the requirement of PLOS ONE, we have submitted the sequences to the NCBI GenBank. The NCBI accession numbers for ERG11 of all the isolates have been provided in Table 2 and S2 Table of the revised manuscript. 

Comment: In your Data Availability statement, you have not specified where the minimal data set underlying the results described in your manuscript can be found. PLOS defines a study's minimal data set as the underlying data used to reach the conclusions drawn in the manuscript and any additional data required to replicate the reported study findings in their entirety. All PLOS journals require that the minimal data set be made fully available.

Response: Without any restriction, we want to make the findings of this study publicly available.

Comment: Upon re-submitting your revised manuscript, please upload your study’s minimal underlying data set as either Supporting Information files or to a stable, public repository and include the relevant URLs, DOIs, or accession numbers within your revised cover letter.

Response: Thank you so much for your suggestion. We have provided the minimal underlying data set as the Supporting Information file in the revised manuscript. We also mentioned in details in the revised cover letter. 

Comment: If there are ethical or legal restrictions to sharing your data publicly, please explain these restrictions in detail. Please see our guidelines for more information on what we consider unacceptable restrictions to publicly.

Response: We don’t have any ethical or legal restrictions for sharing our data. We want to share our data publicly without any restriction.

Comment: We will update your Data Availability statement to reflect the information you provide in your cover letter.

Response: Thank you so much for your assistance.

Comment: PLOS ONE now requires that authors provide the original uncropped and unadjusted images underlying all blot or gel results reported in a submission’s figures or Supporting Information files. When you submit your revised manuscript, please ensure that your figures adhere fully to these guidelines and provide the original underlying images for all blot or gel data reported in your submission.

Response: As per the requirement of PLOS ONE we have provided the original uncropped and unadjusted images of gel results reporting in this manuscript as Supporting Information file in the revised manuscript (S1-S6 Fig). 

Comment: In your cover letter, please note whether your blot/gel image data are in Supporting Information or posted at a public data repository, provide the repository URL if relevant, and provide specific details as to which raw blot/gel images, if any, are not available.

Response: The Supporting Information of this manuscript conation gel image data indicated as S1-S6 Fig. We also noted it in the revised cover letter.

Comment: Thank you for stating the following in the Acknowledgments Section of your manuscript:

'The authors duly acknowledge the Indian Council of Medical Research (ICMR), Government of India for financial supports.'

Please remove any funding-related text from the manuscript and let us know how you would like to update your Funding Statement.

Response: Thank you so much for your valuable suggestion. We removed the funding-related text and relevant changes have been included in the revised manuscript.

Comment: Please include your amended statements within your cover letter; we will change the online submission form on your behalf

Response: Thank you for your assistance. We have included our amended statements in the cover letter.

Comment: PLOS requires an ORCID iD for the corresponding author in Editorial Manager on papers submitted after December 6th, 2016. Please ensure that you have an ORCID iD and that it is validated in Editorial Manager. To do this, go to ‘Update my Information’ (in the upper left-hand corner of the main menu), and click on the Fetch/Validate link next to the ORCID field. This will take you to the ORCID site and allow you to create a new iD or authenticate a pre-existing iD in Editorial Manager.

Response: The ORCID iD for the corresponding author has been updated in the Editorial Manager system 

Reviewers’ comments:    

Reviewer's Responses to Questions:

Comment: Is the manuscript technically sound, and do the data support the conclusions?

Reviewer #1: Partly

Reviewer #2: Yes

Has the statistical analysis been performed appropriately and rigorously?

Reviewer #1: N/A

Reviewer #2: N/A

Have the authors made all data underlying the findings in their manuscript fully available?

Reviewer #1: Yes

Reviewer #2: Yes

Is the manuscript presented in an intelligible fashion and written in standard English?

Reviewer #1: Yes

Reviewer #2: Yes

Response: Thank you so much for your appreciation and positive responses.

Review Comments to the Author:

Reviewer #1: 

Indeed, C. tropicalis is among the most clinically important Candida species and invention of techniques capable of differentiating azole-resistant from azole-susceptible isolates will have clinical implications. Methodologically speaking, however, there are some drawbacks to this study, which are as follows,

Comment: C461T does not confer azole resistance and the inclusion of RSM is not necessary (doi: 10.1016/j.cmi.2018.11.007), since it only detects this mutation.

Response: Although C461T mutation is not directly interfering in drug binding, but several studies including our study showed that both A395T and C461T mutations are appearing together. In our study, all the resistant isolates presented these two mutations. In this context, detection of any of these mutations can confirm the presence of azole resistance in Candida tropicalis isolates. As per our suggestion detection of C461T mutation is also very much important for the rapid detection of resistance. Since both these mutations occur simultaneously, the detection of C461T mutation could be used as a surrogate marker of azole resistance. Therefore, we want to keep the RSM approach for the rapid detection of resistance in the revised manuscript.

Comment: The HRM assay shows a very subtle difference concerning the Tm values of the PCR product obtained from azole-resistant and azole-susceptible isolates, which for sure will cause misidentification and even this slight difference is because of substitution of Cytosine to Thymidine, while the substitution of Adenosine to Thymidine per se will not be differentiated via HRM application. Therefore, these points will also undercut the applicability of HRM and RSM, but ARMS PCR, only for A395T will be important.

Response: Although HRM assay showed small difference in Tm between the resistant and susceptible isolates, the difference is highly significant and sufficient to discriminate the azole resistant and susceptible isolates. The sensitivity and specificity of the HRM approach were significantly higher in our study, which confirms that the chance of misidentification is very low. 

Yes, this slight difference is because of substitution of the Cytosine to Thymidine, not for Adenosine to Thymidine. As we already mentioned that all of our azole resistant isolates showed that both A395T and C461T mutations are appearing together. In this context, detection of any of the mutations is equally significant for the detection of azole resistance in C. tropicalis. 

Comment: In light of these points, authors are encouraged to obtain more azole-resistant C. tropicalis isolates carrying Y132F and azole-susceptible isolates, prepare blinded test sets, and subject them to their ARMS PCR. This will establish the basis for the applicability of ARMS PCR to be used in developing countries, where ICs are mainly treated by azoles and also more advanced PCR machines, such as real-time PCR, might not be that available (doi: 10.3390/jof5040090). Try to use a systematic flow proving the efficacy of ARMS PCR and supplement your findings with ROC curve and well-arranged figures.

Response: Thank you so much for your meticulous comments. All the resistant isolates with ERG11 mutations obtained in between 2015 to 2018 have been included in the present study. We are routinely using these techniques for rapid resistance screening. We are in a process of validating all the approaches with a large number of isolates in near future. In our setup, for the rapid detection of azole resistance ARMS PCR, RSM and HRM approaches are equally helpful and could be implemented in developing countries, where ICs are mainly treated by azoles. In the present study, we showed that the running cost for all three methods are very less and can be implemented in a limited setup. In the present study, the sensitivity and specificity of ARMS PCR are 100%, confirming that the efficiency is also 100%. Therefore, we did not analyze the ROC curve. We are providing more figures with more number of isolates (Fig S2 & S4). 

Comment: Line 59, please note that C. tropicalis is the first common cause of candidemia in Tunisia (doi: 10.3109/13693786.2010.493561) and also Algeria (doi: 10.1186/s13756-020-00710-z). 

Response: Thank you so much for your valuable suggestion. Relevant changes have been included in the revised manuscript. 

Comment: Lines 60-63. Please note that triazoles are more commonly used in developing countries, where the high expenses of echinocandins deter their wide use in such countries. 

Response: Thank you for your suggestion. Required modifications have been included in the revised manuscript. 

Comment: Lines 63-66. Please note that azole-resistance is not caused by host factors rather by the prolonged and/ or previous exposure with azoles and poor hand hygiene and infection control measures may further amplify the azole-resistant isolates in clinical settings. Authors may meant azole therapeutic failure. Also, please highlight the fact that azole-resistant C. tropicalis isolates may even occur in azole-naïve patients that might suggest horizontal transfer in clinical settings. Examples are, doi: 10.1007/s10156-012-0412-9, doi: 10.3201/eid2509.190520, doi.org/10.1093/mmy/myz124.

Response: According to the available literature several host factors like patients with a compromised immune system, indwelling catheters, artificial heart valves, and other implanted devices are associated with drug resistant ICs. Additionally, the drug penetration at sites of infection is poorly understood (doi:10.1016/S1473-3099(17)30316-X). As per your suggestion regarding horizontal transfer in clinical settings, relevant modifications have been included in the revised manuscript. 

Comment: As mentioned above, please note that C461T does not confer azole resistance, which has been proved via heterologous expression analysis (doi: 10.1016/j.cmi.2018.11.007). 

Response: Required modification has been included in the revised manuscript.

Comment: Lines 73-79. These statements are too strong and authors are encouraged to modify this, since there are some studies also have shown the otherwise (doi.org/10.1093/mmy/myz124) and not all azole-resistant isolates harbor accountable mutations in ERG11. Indeed, azole resistance, unlike echinocandin resistance, involves numerous players and their contribution in concert will cause azole resistance. Moreover, here authors have just focused on one mutation, Y132F, while other mutations, although with a lower prevalence, confer azole resistance, such as G464S/D, Y125F, P56S, etc.

Response: Thank you so much for your meticulous comment. Relevant changes have been included in the revised manuscript. 

Comment: It seems unclear why authors developed so many methods, while they could have focused on one method. I mean, this is not reflected in introduction.

Response: In the present study our aim was the rapid detection of ERG11 mutations for which we standardized three approaches for a comprehensive evaluation, instead of developing one approach, since multiple approaches can cross validate the results. We found that all three approaches are equally efficient and the running cost for all three methods is very low 

Comment: Lines 108-115, please delete them and simply refer to a study. No need for such elaboration on DNA extraction. 

Response: As per your suggestion, we have deleted this portion in the revised manuscript.

Comment: Why authors used such huge quantity of DNA samples, 100ng DNA? Even using 1ng DNA any PCR reactions work perfect. 

Response: In the present study, we showed the Analytical sensitivity for all three approaches. We got the best result with 100 ng of DNA and kept it as a standard for all the methods. Although 1ng of DNA is sufficient for PCR amplification, for ARMS PCR at least 10 ng DNA input is needed for proper detection. Therefore, we selected 100 ng of DNA as a standard to get the finest result. 

Reviewer #2: 

Comment: None. Please see my comments in the attachment.

And I have no competing interest that interferes with or could be perceived as potentially interfering with, a thorough and objective assessment of this manuscript. 

Response: Thank you so much for reviewing our manuscript and for your positive responses. 

Comment: PLOS authors have the option to publish the peer review history of their article. If published, this will include your full peer review and any attached files.

Response: We have no problem to publish the peer review history of their article.

---

## [Decision Letter · Decision Letter 1]

30 Oct 2020

PONE-D-20-17635R1

Rapid detection of ERG11 polymorphism associated azole resistance in Candida tropicalis

PLOS ONE

Dear Dr. Ghosh,

Thank you for submitting your manuscript to PLOS ONE. After careful consideration, we feel that it has merit but does not fully meet PLOS ONE’s publication criteria as it currently stands. Therefore, we invite you to submit a revised version of the manuscript that addresses the points raised during the review process.

Thank you for addressing the concerns of the reviewers. Please consider and offer a response to Reviewer #1. If possible,  address this issue in a short comment in discussion of limitations of your study.

We look forward to receiving your revised manuscript.

Kind regards,

Joy Sturtevant

Academic Editor

PLOS ONE

Reviewers' comments:

Reviewer's Responses to Questions

**Comments to the Author**

1. If the authors have adequately addressed your comments raised in a previous round of review and you feel that this manuscript is now acceptable for publication, you may indicate that here to bypass the “Comments to the Author” section, enter your conflict of interest statement in the “Confidential to Editor” section, and submit your "Accept" recommendation.

Reviewer #1: All comments have been addressed

2. Is the manuscript technically sound, and do the data support the conclusions?

Reviewer #1: Partly

3. Has the statistical analysis been performed appropriately and rigorously? 

Reviewer #1: N/A

4. Have the authors made all data underlying the findings in their manuscript fully available?

Reviewer #1: Yes

5. Is the manuscript presented in an intelligible fashion and written in standard English?

Reviewer #1: Yes

6. Review Comments to the Author

Reviewer #1: Clearly authors have tried to address some of the issues with the paper and overall the paper is improved. I am afraid that I do not agree with the author’s statement that Y132F is always accompanied by C461T. Indeed, there are numerous cases in which Y132F (A395T) does occurs alone without C461T. As suggested before, this will create a significant challenge to the HRM technique, since a) Y132F is resulted from substitution of A to T, therefore it will not have that much influence on the melting temperature of the amplicons with only Y132F, and b) the fact that Y132F can occur alone preclude the usage of this technique as a comprehensive surrogate marker. Therefore, the collection of fluconazole-resistant C. tropicalis isolates used by authors will not dictate what can be found in other centers, https://www.mdpi.com/2309-608X/6/3/138. Moreover, in our laboratory we have used a lot of times to adjust the TM difference between susceptible and resistant isolates as far as possible (at least more than one degree of centigrade) to prevent misidentification and we have witnessed cases, where amplicons with a difference of one degree of centigrade were misidentified when tested in the other labs, which is inherent and may arise due to change in setting, reagent, the DNA extraction method, the purity of the DNA samples, etc. Finally, the TM difference authors observed is just a fraction of centigrade and it can cause misidentification in isolates carrying A395T+C461T and will not differentiate those only carrying A395T. The same concern is also applicable to restriction site mutation approach, since it only cut C461T and cannot differentiate resistant isolates carrying only Y132F from susceptible isolates. It is understandable that authors spent time, energy, and money to develop these assays, but the final outcome of the technique should be considered. If authors still advocate their statements, they are encouraged to mention these concerns as the pitfall of their techniques in discussion and call for improvements.

Our goal as a reviewer is to enhance the quality of the papers and do hope that authors comprehend that.

7. PLOS authors have the option to publish the peer review history of their article (what does this mean?). If published, this will include your full peer review and any attached files.

Reviewer #1: No

---

## [Author Response · Author response to Decision Letter 1]

5 Dec 2020

Editor comments:

Comment: Please consider and offer a response to Reviewer #1. If possible, address this issue in a short comment in discussion of limitations of your study.

Response: Thank you for considering our manuscript in your esteemed journal. We have tried our best to address all the issues raised by the Reviewer #1. We have addressed this issue in a short comment in discussion as the limitations of our study in the revised manuscript. 

Comment: Please include the following items when submitting your revised manuscript:

Response: We are submitting the rebuttal letter containing each point raised by the academic editor and reviewers.

We are also sending one copy of the ‘Revised Manuscript with Track Changes’ and one clean copy of the revised version labeled as ‘Manuscript’. 

Comment: Response: Required modifications in the financial disclosure has included in the updated statement of the cover letter. 

Comment: Guidelines for resubmitting your figure files are available below the reviewer comments at the end of this letter.

Response: We are resubmitting the figure files according to the guidelines of this journal. 

Reviewers’ comments:    

Reviewer's Responses to Questions:

Comment: 1. If the authors have adequately addressed your comments raised in a previous round of review and you feel that this manuscript is now acceptable for publication, you may indicate that here to bypass the “Comments to the Author” section, enter your conflict of interest statement in the “Confidential to Editor” section, and submit your "Accept" recommendation.

Reviewer #1: All comments have been addressed

2. Is the manuscript technically sound, and do the data support the conclusions?

Reviewer #1: Partly

3. Has the statistical analysis been performed appropriately and rigorously?

Reviewer #1: N/A

4. Have the authors made all data underlying the findings in their manuscript fully available?

Reviewer #1: Yes

5. Is the manuscript presented in an intelligible fashion and written in standard English?

Reviewer #1: Yes

Response: Thank you so much for your appreciation and positive responses.

Review Comments to the Author:

Reviewer #1: 

Clearly authors have tried to address some of the issues with the paper and overall the paper is improved. I am afraid that I do not agree with the author’s statement that Y132F is always accompanied by C461T. Indeed, there are numerous cases in which Y132F (A395T) does occurs alone without C461T. As suggested before, this will create a significant challenge to the HRM technique, since a) Y132F is resulted from substitution of A to T, therefore it will not have that much influence on the melting temperature of the amplicons with only Y132F, and b) the fact that Y132F can occur alone preclude the usage of this technique as a comprehensive surrogate marker. Therefore, the collection of fluconazole-resistant C. tropicalis isolates used by authors will not dictate what can be found in other centers, https://www.mdpi.com/2309-608X/6/3/138. Moreover, in our laboratory we have used a lot of times to adjust the TM difference between susceptible and resistant isolates as far as possible (at least more than one degree of centigrade) to prevent misidentification and we have witnessed cases, where amplicons with a difference of one degree of centigrade were misidentified when tested in the other labs, which is inherent and may arise due to change in setting, reagent, the DNA extraction method, the purity of the DNA samples, etc. Finally, the TM difference authors observed is just a fraction of centigrade and it can cause misidentification in isolates carrying A395T+C461T and will not differentiate those only carrying A395T. The same concern is also applicable to restriction site mutation approach, since it only cut C461T and cannot differentiate resistant isolates carrying only Y132F from susceptible isolates. It is understandable that authors spent time, energy, and money to develop these assays, but the final outcome of the technique should be considered. If authors still advocate their statements, they are encouraged to mention these concerns as the pitfall of their techniques in discussion and call for improvements.

Our goal as a reviewer is to enhance the quality of the papers and do hope that authors comprehend that.

Response: Thank you so much for your meticulous comment and valuable suggestion. As per your suggestion, we have mention these concerns as the pitfall of our techniques in discussion of the revised manuscript. 

Comment: PLOS authors have the option to publish the peer review history of their article. If published, this will include your full peer review and any attached files.

Response: We have no problem to publish the peer review history of our article.

We would like to thank the Academic Editor and Reviewers for their important comments and valuable suggestions that have been improved the manuscript significantly.

---

## [Editor Report · Decision Letter 2]

23 Dec 2020

Rapid detection of ERG11 polymorphism associated azole resistance in Candida tropicalis

PONE-D-20-17635R2

Dear Dr. Ghosh,

We’re pleased to inform you that your manuscript has been judged scientifically suitable for publication and will be formally accepted for publication once it meets all outstanding technical requirements.

Kind regards,

Joy Sturtevant

Academic Editor

PLOS ONE
---

## [Editor Report · Acceptance letter]

2 Jan 2021

PONE-D-20-17635R2 

Rapid detection of *ERG11* polymorphism associated azole resistance in *Candida tropicalis*

Dear Dr. Ghosh:

I'm pleased to inform you that your manuscript has been deemed suitable for publication in PLOS ONE. Congratulations! Your manuscript is now with our production department. 

Kind regards, 

on behalf of

Dr. Joy Sturtevant 

Academic Editor

PLOS ONE